# 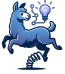 AdaReasoner: Adaptive Reasoning Enables More Flexible Thinking

**Xiangqi Wang**[1]* **Yue Huang**[1]* **Yanbo Wang**[2] **Xiaonan Luo**[1] **Kehan Guo**[1]
**Yujun Zhou**[1] **Xiangliang Zhang**[1]†
[1] University of Notre Dame [2]MBZUAI

{xwang76, yhuang37, xluo6, kguo2, yzhou25, xzhang33}@nd.edu
yanbo.wang@mbzuai.ac.ae

## Abstract

LLMs often need effective configurations, like temperature and reasoning steps, to handle tasks requiring sophisticated reasoning and problem-solving, ranging from joke generation to mathematical reasoning. Existing prompting approaches usually adopt general-purpose, fixed configurations that work "well enough" across tasks but seldom achieve task-specific optimality. To address this gap, we introduce AdaReasoner, an LLM-agnostic plugin designed for any LLM to automate adaptive reasoning configurations for tasks requiring different types of thinking. AdaReasoner is trained using a reinforcement learning (RL) framework, combining a factorized action space with a targeted exploration strategy, along with a pretrained reward model to optimize the policy model for reasoning configurations with only a few-shot guide. AdaReasoner is backed by theoretical guarantees and experiments of fast convergence and a sublinear policy gap. Across six different LLMs and a variety of reasoning tasks, it consistently outperforms standard baselines, preserves out-of-distribution robustness, and yield gains on knowledge-intensive tasks through tailored prompts. Introduction of this paper can also be viewed publicly at https://mine-lab-nd.github.io/project/adareasoner.html.

## 1 Introduction

Large Language Models (LLMs) have achieved impressive advancements across a wide range of natural language processing tasks, including syntactic parsing [26], complex scientific reasoning [52], and commonsense knowledge answering [59]. As the model size and training data scale up, LLMs have demonstrated the ability to surpass human-level accuracy on certain benchmarks [45], highlighting their emerging capacity for sophisticated reasoning and problem-solving.

To better enhance LLM reasoning capabilities–and to push their performance closer to, or even beyond, human-level reasoning–numerous prompting-based strategies have been proposed. Chain-of-Thought (CoT) prompting encourages explicit decomposition of complex problems into intermediate steps [54, 62], while Tree-of-Thought (ToT) generalizes this idea by exploring multiple branching reasoning paths [57]. Sampling-based approaches like Best-of-N improve robustness by selecting the most coherent reasoning path from diverse candidates [16], and automatic prompt optimization techniques aim to systematically discover prompts that better facilitate multi-step reasoning [58, 42]. If samples of the same type of question are provided, In-Context Learning (ICL) [5] also prompts LLM with few-shot examples with advanced performance.

Despite these advances, LLM reasoning remains highly configuration-sensitive: as Figure 1 shows, GPT-4o's accuracy on the metaphor expression classification task [49] swings wildly under different

---

*Equal contribution.

†Corresponding author: xzhang33@nd.edu

reasoning configurations. While divergent reasoning prompts and fewer reasoning steps could greatly improve performance, temperature as 1 instead drown out useful reasoning with noise, negating any benefit from the added randomness. However, previous methods have not targeted tuning on these parameters. CoT [54, 62] and ToT [57] apply fixed reasoning structures that fail to generalize to creative or subjective tasks (e.g. spatial planning [46]). Best-of-N [16] rely on unguided generation, suffering from a "garbage in, garbage out" effect. Automatic prompt optimization [58, 42] focuses on static templates and overlooks crucial hyperparameters like temperature, failing to adjust reasoning strategies. While ICL [5] extracts some cues from input questions, it remains brittle under context perturbations [28], and its reliance on implicit pattern matching has been shown to be less effective than direct structured reasoning [48]. These limitations call a need for an adaptive prompting configuration strategy for LLMs to handle various sophisticated reasoning.

However, identification of the optimal prompting configuration for LLMs is a non-trivial task. First, task types span logical, creative, and subjective domains, often in combination, so that many queries cannot be neatly categorized or matched with pre-set configurations template. This necessitates strategies that are highly adaptive and tailored to the specific demands of each question. Second, LLM reasoning capability is sensitive to the configuration settings that involve multiple factors, as shown in Figure 1. The search space spanned by these factors when selecting effective configurations is combinatorially large. This presents a challenge for building a decision-making model that tailors the configuration for each input task. Third, while building such a model using a data-driven approach is promising, exhaustively collecting training samples for every possible

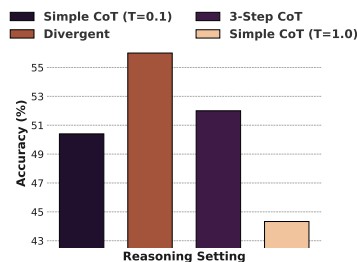

Figure 1: Performance of different CoT settings on the metaphor dataset [49]. The default temperature is 0.1 if not specified.

configuration is computationally expensive and impractical. This necessitates an approach that can generalize from a limited set of examples and capture reasoning patterns that are transferable across similar tasks.

We introduce AdaReasoner, an LLM-agnostic plugin designed to automate adaptive reasoning configurations for tasks requiring diverse types of thinking. When integrated with an LLM, AdaReasoner is trained using a reinforcement learning (RL) framework. In this setup, AdaReasoner acts as a decision-making agent, where the state is defined by the current task presented to the LLM, reflecting the nature of the reasoning required (e.g., logical, creative, or subjective). The action corresponds to selecting a configuration from an action space composed of three key hyperparameters: (i) the reasoning instruction formats, (ii) the generation temperature, and (iii) the number of reasoning steps. To enable AdaReasoner to learn the most effective configuration policy, a pretrained reward model is employed to evaluate the effectiveness of the reasoning configuration. This model provides feedback to guide the agent's learning, enabling it to efficiently acquire effective configurations with only limited guidance (i.e., few-shot learning). To facilitate exploration and improve generalization, we employ a Boltzmann exploration mechanism, enabling the agent to explore and optimize configurations more effectively during training. Once trained, AdaReasoner is used as a plug-in to the LLM, providing adaptive reasoning configurations that allow the model to adjust its reasoning approach based on the task at hand.

Our contributions can be summarized as the followings:

- We introduce AdaReasoner, an LLM-agnostic plugin that automates adaptive reasoning configurations for tasks requiring diverse types of thinking.
- AdaReasoner leverages a reinforcement learning framework with a factorized action space. Its training is data-efficient yet scalable, requiring only a small number of samples for each task aided by the use of the Boltzmann exploration mechanism.
- Extensive evaluations on diverse tasks show that AdaReasoner outperforms standard CoT and baselines, and sustains strong OOD performance.

## 2 Related Work of Reasoning in LLMs

The pursuit of enhanced reasoning capabilities in LLMs has spurred diverse research trajectories, beginning with foundational techniques like Chain-of-Thought (CoT) prompting [54]. CoT enables LLMs to articulate intermediate steps, significantly improving performance on complex tasks. However, its efficacy can be hampered by sensitivity to prompt formulation [44, 33] and limitations in

subjective or creative domains [7, 56], sometimes even degrading performance where brevity is key [24]. To mitigate these issues and reduce manual effort, innovations such as Automatic CoT (Auto-CoT) [58, 42] emerged, automating the generation of effective reasoning exemplars. Further advancements include structured reasoning frameworks like Tree-of-Thoughts (ToT) [57] and Graph-of-Thoughts (GoT) [4], which allow models to explore and evaluate multiple reasoning pathways, alongside methods like CoT-influx [13] that optimize few-shot CoT contexts.

To bolster the robustness and reliability of LLM reasoning, researchers have explored self-correction and learning-based paradigms. Self-consistency techniques [53], often realized through Best-of-N sampling, leverage the generation of multiple diverse reasoning paths and subsequent aggregation (e.g., via majority voting) to improve answer accuracy. Complementary to this, self-reflection mechanisms, as seen in Self-Refine [27] and Reflexion [41], empower LLMs to iteratively critique and enhance their own outputs, akin to human error correction, with some approaches fine-tuning with divergent CoT to specifically boost these capabilities [33]. Reinforcement Learning (RL) has also become a cornerstone for optimizing reasoning, from general alignment via RLHF [31] to specialized reward models that guide the LLM towards more accurate and effective thought processes [15]. Models like DeepSeek-R1 [11] exemplify LLMs fine-tuned with RL to excel at intricate reasoning, sometimes learning to control their own reasoning flow through meta-actions.

The nuanced control of generation parameters and adaptive hyperparameter tuning represent another critical frontier. The stochastic decoding settings, such as temperature, significantly affect output diversity and, consequently, reasoning quality and creativity [35]. Higher diversity can fuel methods like self-consistency but requires careful management to maintain coherence. Recent work has thus focused on automated optimization of prompt configurations, decoding parameters, and even enabling LLMs to self-regulate their generation strategies, as demonstrated by Hyperparameter-Aware Generation (HAG) [51]. Our AdaReasoner contributes to this line of research by introducing an adaptive framework that explicitly manages a toolbox of reasoning hyperparameters, including the reasoning method prompt, temperature, and number of reasoning steps, using an RL-trained agent to dynamically tailor the reasoning process to individual inputs, coupled with self-reflection and a robust selection mechanism for enhanced flexibility.

## 3  AdaReasoner

**Motivation.** Even though CoT and similar LLM reasoning methods have been studied as generally efficient and helpful, they still cannot achieve ideal performance across all types of questions. For example, tasks like joke generation or metaphor interpretation often require divergent and creative reasoning chain [61]. For more complex reasoning tasks, stronger and more explicit reasoning instructions would be beneficial [22]. Thus, adapting LLM configurations tailored for specific tasks is crucial for achieving better overall performance. As illustrated in Figure 2, we design AdaReasoner to adapt reasoning configurations by taking actions as a combination of different hyperparameters for LLMs. The inference/evaluation process is illustrated by the black arrows, while the training flow is depicted by the cyan arrows.

**Problem Formulation.** The goal of AdaReasoner is to adaptively find the most effective hyper-parameter configuration $a$ for a given question $q$ such that an LLM (denoted as $\Phi$) generates the correct reasoning answer $\Phi(q|a)$. More specifically, the configuration $a$ is a 3-dimensional vector, where each element corresponds to one of the three hyperparameters: $a_t$ (generation temperature), $a_p$ (reasoning instruction format), and $a_s$ (the number of reasoning steps). Denoting AdaReasoner as $\Pi_\Theta$, our goal is to train its neural network weights $\Theta$ to learn the optimal policy for deciding the configuration $a$ given a question $q$. By considering the question $q$ along with the LLM $\Phi$ as the state, the decision-making process is represented as $a = \Pi_\Theta(q, \Phi)$. During training, we employ a pre-trained model (e.g. DeBERTa in huggingface) as the reward model $r$ to provide feedback on the generated answer by comparing it to the ground truth reference $R$ from the training data, i.e., $r(\Phi(q|a), R)$. In this approach, we address the issue that it is not possible to directly evaluate the quality of generated configuration $a$, as there is no ground truth for $a$ itself. Instead, the effectiveness of $a$ is judged indirectly based on the resulting answer $\Phi(q|a)$, ensuring that the AdaReasoner agent is informed about the quality of its reasoning configuration through the answer's relevance and accuracy.

Within the broader RL framework, our study can be viewed as a *multi-armed bandit* problem, where the *arms* represent different configuration actions. Each question is an independent task (state),

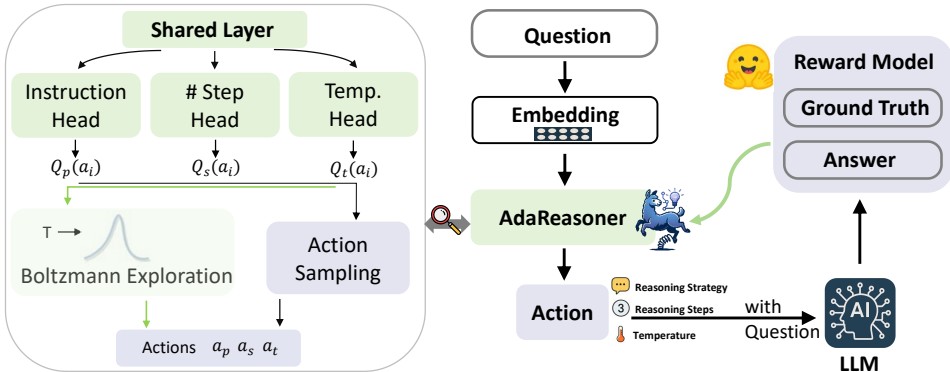

Figure 2: The proposed framework of using AdaReasoner for automating the reasoning configurations (instructions, steps, temperature). During training, configurations actions are sampled with Boltzmann exploration, guiding LLMs to generate answers, which are then evaluated by a reward model for policy optimization.

where the agent determines the actions (sets the values for all arms), receives a reward based on the effectiveness of the answer, and then moves on to the next task. The objective is to optimize the selection of hyperparameters to maximize the reward for each question. Therefore, given a set of training questions and reference answer samples $\mathcal{D}_{\text{train}} = \{(q_i, R_i)\}_{i=1}^{M}$, the objective is to train the AdaReasoner agent as

$$\Theta^* = \arg\max_{\Theta} \mathbb{E}_{(q,R)\sim\mathcal{D}_{\text{train}}} \mathbb{E}_{a\sim\Pi_\Theta(a|q,\Phi)} \Big[ r\Big(\Phi(q|a), R\Big)\Big]. \tag{1}$$

Theoretical analysis about AdaReasoner is presented in Appendix B, with a step-by-step description in Algorithm 1.

## 3.1 Hyperparameter Configuration (Action)

As mentioned earlier, we consider three hyperparameters in the reasoning configuration: 1) the generation temperature $a_t$; 2) the format of reasoning instructions $a_p$; and 3) the number of reasoning steps in CoT $a_s$, for several reasons. First, they have substantial impacts on the reasoning performance. Previous studies have revealed that the generation temperature modulates the diversity of model outputs, often yielding markedly different responses when varied [36]. The number of reasoning steps reflects the depth and thoroughness of the inference process and it thus could influence the reasoning accuracy [9, 17]. The format of reasoning instructions, such as backward reasoning and step-by-step deduction, also plays a crucial role in guiding the model's reasoning process [2, 50]. Second, the settings of these three hyperparameters are adaptable for both proprietary and open-weight LLMs, with enhancement of adareasoner's versatility. Third, we are aware of other hyperparameters that may also impact reasoning, such as the $p$ in top-$p$ sampling during generation and the random seed. However, we exclude $p$ because tuning top-$p$ alongside temperature is not recommended together with temperature [1]. Additionally, our empirical evaluation found that varying the random seed could not be beneficial for improving LLMs' reasoning performance (as shown in Section 4.3).

To ensure practical feasibility, these configuration actions are discretized with a finite set of options. Specifically, the number of reasoning steps is bounded to avoid extreme values, defined as $\mathcal{A}_s$ as integers set, and temperature is discretized as set $\mathcal{A}_t$. The options for reasoning instructions, denoted as $\mathcal{A}_p$, are constructed based on a compositional design grounded in structure-mapping theory from cognitive psychology [10], which models human reasoning by composing a **core** reasoning structure with **contextual** modifications. Accordingly, each reasoning instruction is factorized into two components: a **base** component, which specifies the overall cognitive strategy (e.g., creative thinking, analogical mapping, self-audit [6]), and a **variation**, which modulates the emphasis on specific parts of the question or modifies the reasoning surface form. For example, a **base** *"Apply creative reasoning to unearth unconventional insights and challenge standard assumptions"* could be combined with a **variation** *"Use simple, straightforward language to guarantee clarity and accessibility"* for guiding the reasoning of divergent thinking types of problems. The same **base**, when combined with a **variation** *Validate conclusions by aligning them with established principles or empirical data*, such instruction is useful for critical thinking types of reasoning problems. Detailed of

the base and variation components and their instantiation are provided in Appendix C. The reasoning instruction action space, $\mathcal{A}_p$, is composed of pairs in the form of {base, variation}. Each action $a_p$ corresponds to one of the possible combinations of a base and its associated variation.

Ultimately, the decision about the action involves selecting a generation temperature $a_t$ from $\mathcal{A}_t$, a number of reasoning steps $a_s$ from $\mathcal{A}_s$, and one form of reasoning instruction $a_p$ from $\mathcal{A}_p$.

### 3.2 Design and Training of AdaReasoner

**Neural Architecture of AdaReasoner.** As shown in Figure 2, the input query question, after embedding, undergoes three action selections before being sent to the LLMs for reasoning to generate the answer. While the embedding is performed (e.g. by pre-trained BERT model [55]), the trainable neural network parameters of AdaReason consist of three parallel channels, each corresponding to one action, and one shared common layer as in Figure 2. The workflow is as follows: let $Embed(q)$ be the embedding of the input question $q$. It is first passed through the common layer to obtain $h = f_{\theta_c}(Embed(q))$, where $\theta_c$ are the parameters of the common layer (e.g., a fully connected MLP), and $h$ captures the features necessary to determine the actions.

Then $h$ is sent to each channel, where the action selection is performed as

$$a_p \sim \pi_p(\cdot|h) = f_{\theta_p}(h), \quad a_t \sim \pi_t(\cdot|h) = f_{\theta_t}(h), \quad a_s \sim \pi_s(\cdot|h) = f_{\theta_s}(h), \tag{2}$$

where each policy $\pi(\cdot|h)$ is implemented as a feed-forward network.

This design factorizes the policy $\Pi$ into three independent heads, each handling a specific action selection, significantly reducing optimization space from multiply to summary. Viewing $\Pi$ as multi-armed bandit problem, it is factorizing the joint-arms into set of parallel yet not independent single arm ones. While each head operates independently, they are optimized jointly with a shared latent representation, ensuring coherent decision-making and unified optimization across $a_p$, $a_s$ and $a_t$. Let $K = MT$ be the total number of steps in learning, where $M$ is the number of training questions and $T$ is the number of trials for each question. We analyze the regret of AdaReasoner, i.e., the reward difference between AdaReasoner and the optimal policy without factorization in App. B. The regret per step is bounded by $O\left(\left(\frac{|\mathcal{A}| \ln |\mathcal{A}|}{K}\right)^{0.5}\right)$, where $|\mathcal{A}|$ is the total number of action values: $\mathcal{A} = \mathcal{A}_p \times \mathcal{A}_t \times \mathcal{A}_s$. This shows that the regret per step becomes negligible once $K \gg |\mathcal{A}| \ln |\mathcal{A}|$, which is consistent with the empirical observation of few-shot convergence, meaning that AdaReasoner learns effectively with relatively few training examples. Moreover, under Lipschitz smoothness and bounded variance conditions, Adareasoner with $J(\Theta^*)$ denoted as optimal expected-reward objective and $J(\Theta_0)$ as initial objective achieves an error bound $\frac{2\left(J(\Theta^*) - J(\Theta_0)\right)}{\eta K} + L \eta \sigma^2$ (App. B), reinforcing rapid convergence in the few-shot setting.

**Exploration Strategy.** By formulating the configuration selection for each question as a multi-armed bandit (MAB) problem, we aim to design an effective exploration strategy under the few-shot training setting. However, since the reward is derived indirectly from LLM outputs and the process is not an online learning scenario, standard MAB strategies such as Upper Confidence Bound (UCB) [47] become impractical. Moreover, evaluating all configurations for each context $q$ is computationally infeasible, especially given the noisy and implicit reward landscape induced by LLM responses. Therefore, it is crucial to explore broadly across the configuration space while still prioritizing high-reward actions, and Boltzmann exploration offers an effective solution [32], as it allows the agent to probabilistically select actions based on their estimated rewards. Specifically, for each action ($a_t$, $a_s$ or $a_p$), we estimate the selection probability for its all possible values (in $\mathcal{A}_t$, $\mathcal{A}_s$ or $\mathcal{A}_p$),

$$P(a_i) = \frac{\exp\left(Q(a_i)/\tau\right)}{\sum_{a_j \in \mathcal{A}} \exp\left(Q(a_j)/\tau\right)}, \tag{3}$$

where $Q(a_i)$ is the logit score in the output layer of one policy network $f_\theta$ for action $a_i$. The temperature $\tau$ in Boltzmann exploration controls the exploration-exploitation trade-off: higher $\tau$ promotes exploration, lower $\tau$ favors exploitation. We anneal $\tau$ exponentially as $\tau_t = \tau_0 \cdot \alpha^t, t \leq T$, allowing the policy to gradually shift from broad exploration to reliable configuration selection and refined optimization [19].

**Reward Signal.** Similar to previous work [20, 25, 37] using pre-trained language model as reward on light-weight RL model, we employ a language judgement model (ours is DeBERTa-based) as

reward model [30] to provide feedback on the selected actions. Concretely, for the resulting generated answer $\Phi(q|a)$, it is presented to the reward model alongside the original question $q$ and reference answer $R$ in the form of the prompt "For $q$, the generated answer $\Phi(q|a)$ matches the ground truth $R$ and is correct". The reward is computed from the model's logits, providing a scalar score that enables fine-grained, differentiable supervision over diverse reasoning trajectories.

With the reward $r$, the AdaReasoner is optimized using the gradient descent (REINFORCE) algorithm [43], where the overall policy $\Pi_\Theta(a \mid q, \Phi)$ is factorized into three heads with a shared feature extractor $f_{\theta_c}$, and $\Theta = \{\theta_c, \theta_p, \theta_t, \theta_s\}$ denotes the complete set of trainable parameters. For each head $j \in \{p, t, s\}$, we define the head-specific loss as $\mathcal{L}_j = -r \log \Pi_{\theta_j}(a \mid q, \Phi)$, resulting in a total loss $\mathcal{L} = \sum_{j \in \{p,t,s\}} \mathcal{L}_j$. The gradients are then computed via the chain rule, where the shared-layer gradient is aggregated as $\nabla_{\theta_c} \mathcal{L} = \sum_{j \in \{p,t,s\}} \nabla_{\theta_c} \mathcal{L}_j$, and used for updating

$$\theta_c \leftarrow \theta_c - \eta \, \nabla_{\theta_c} \mathcal{L}. \tag{4}$$

Each head is updated as

$$\theta_j \leftarrow \theta_j - \eta \, \nabla_{\theta_j} \mathcal{L}_j \quad \forall \quad j \in \{p, t, s\}. \tag{5}$$

This training scheme ensures that each sub-policy is guided by its own loss while the shared feature extractor $f_{\theta_c}$ is jointly optimized by all heads, thereby promoting coherence across the three action dimensions and preventing convergence to conflicting optima, in line with findings from multi-task learning [38]. Further training details are described in Algorithm 1.

## 4 Experiments

### 4.1 Experimental Setting

**Dataset.** To evaluate the performance of AdaReasoner, we selected datasets that engage distinct cognitive processes, ranging from logical and mathematical to figurative and generative reasoning.

- **MMLU:** This is a collection of data examples that are in the *Math* category from the Massive Multitask Language Understanding (MMLU) benchmark [12], focusing on numerical reasoning, symbolic manipulation, and procedural problem solving.
- **Metaphor** [49]: This dataset focuses on evaluating whether a highlighted word in context is used metaphorically in the context.
- **TruthfulQA** [21]: This dataset tests LLM trustworthy generation by posing questions with common misconceptions or false premises.
- **LogiQA** [23]: This dataset is designed for multi-step logical reasoning based on Chinese civil service exam questions.

Each dataset contributes 250 samples, randomly sampled from the full dataset. The combined dataset is then divided into a training set of 100 samples and a test set of 900 samples forming thus a few-shot setting. Examples of the four datasets are displayed at Table 5 and distribution of each dataset is shown at Figure 5.

**Baselines.** We compare AdaReasoner with several baselines that adopt different strategies to improve LLM reasoning:

- **CoT (Chain-of-Thought)** [54]: Prompts the model to think step-by-step for reasoning.
- **Think Short**: Prompts the model for brief, quick responses with prompt at Figure 10.
- **ToT (Tree-of-Thought)** [57]: Structures reasoning path as a tree, exploring and selecting among multiple paths.
- **Best-of-N** [16]: Produces $N$ candidate chains, selects the best based on a predefined scoring metric.
- **Auto-CoT** [58]: For each query, retrieve semantically nearest exemplars from a few-shot pool (via embedding clustering), generate CoT rationales, and concatenate the question–rationale–answer triplets as the in-context prompt; other settings follow the original.
- **In-context CoT (ICL)** [5]: Leverages in-context CoT generation by presenting examples of few-shot train set directly within the prompt.

**Evaluation and other details.** To evaluate the alignment between LLM-generated responses and the ground truth, we adopt the "LLM-as-a-Judge" paradigm [60], utilizing GPT-4o to assess both the semantic equivalence of answers and the quality of their explanations through dedicated judgment prompts, as illustrated in Figure 8. In each evaluation, the `top_p` parameter is set to 0.1 and the `max_token` parameter is set to 5,000, with no system prompt utilized. We random select 100 out of

Table 1: Performance of various reasoning methods across multiple datasets for different LLM models (accuracy in %). The highest score for each dataset and the average in each model group is highlighted in **bold** and underlined.

| Model | Reason Method | Dataset (%) | | | | Average |
| | | Metaphor | TruthfulQA | MMLU | LogiQA | |
|---|---|---|---|---|---|---|
| **GPT-4o** | CoT | 50.40 | 78.40 | 76.04 | 70.00 | 68.71 |
| | Think Short | 61.00 | 64.81 | 68.52 | 70.81 | 66.28 |
| | ToT | 48.25 | 74.29 | 86.11 | 73.90 | 70.91 |
| | Best-of-N | 52.60 | 79.41 | 83.41 | 72.37 | 71.95 |
| | Auto-CoT | 62.33 | **83.09** | 72.15 | 71.71 | 72.32 |
| | In-context CoT | 53.98 | 77.04 | 83.63 | 80.04 | 74.42 |
| | AdaReasoner | **71.56** | 81.30 | **86.49** | **82.31** | **80.42** |
| **Llama-3.3-70B-Ins.** | CoT | 51.56 | 75.77 | 83.33 | 75.56 | 71.56 |
| | Think Short | 59.56 | 75.77 | 81.61 | 73.78 | 72.68 |
| | ToT | 60.89 | 75.33 | 86.24 | 83.56 | 76.51 |
| | Best-of-N | 52.89 | 77.09 | **89.69** | 76.00 | 73.92 |
| | Auto-CoT | 45.33 | 78.85 | 81.82 | 76.00 | 70.50 |
| | In-context CoT | 52.71 | 82.45 | 84.57 | 75.59 | 73.60 |
| | AdaReasoner | **66.11** | **83.09** | 87.77 | **85.00** | **80.74** |
| **Qwen-2.5-72B-Ins.** | CoT | 60.18 | 79.36 | 73.89 | 78.26 | 72.92 |
| | Think Short | 71.24 | 80.28 | 64.16 | 75.22 | 72.73 |
| | ToT | 62.26 | 77.50 | 66.57 | 79.51 | 71.46 |
| | Best-of-N | 59.73 | 78.44 | 76.11 | 78.26 | 73.14 |
| | Auto-CoT | 65.93 | 83.49 | 76.11 | 79.13 | 76.17 |
| | In-context CoT | **73.39** | 78.94 | 71.93 | 74.83 | 74.77 |
| | AdaReasoner | 65.19 | **83.82** | **80.14** | **80.79** | **77.49** |
| **Claude-3.5-sonnet** | CoT | 62.13 | 86.13 | 85.00 | 80.43 | 78.42 |
| | Think Short | **67.71** | 83.43 | 78.95 | 77.95 | 77.01 |
| | ToT | 59.45 | 85.12 | 86.43 | 81.98 | 78.25 |
| | Best-of-N | 41.41 | 83.43 | 81.87 | 78.95 | 71.42 |
| | Auto-CoT | 65.04 | 84.86 | 88.50 | 78.70 | 79.28 |
| | In-context CoT | 55.81 | **88.60** | 79.23 | 79.53 | 75.79 |
| | AdaReasoner | 65.77 | 86.17 | **89.21** | **84.55** | **81.43** |
| **Deepseek-R1** | CoT | 54.35 | 83.34 | 96.13 | 81.82 | 78.91 |
| | Think Short | 67.71 | 80.00 | 95.55 | 77.71 | 80.24 |
| | ToT | 63.33 | 86.16 | **98.70** | 83.22 | 82.85 |
| | Best-of-N | 54.55 | 85.51 | 94.37 | 87.01 | 80.36 |
| | Auto-CoT | 61.04 | 82.61 | 97.70 | 80.52 | 80.47 |
| | In-context CoT | 50.06 | 84.21 | 96.15 | 84.25 | 78.67 |
| | AdaReasoner | **72.00** | **88.17** | 96.33 | **88.60** | **86.28** |
| **GPT-o3-mini** | CoT | 45.10 | 84.00 | 95.71 | 83.87 | 77.17 |
| | Think Short | 57.14 | 80.00 | 93.21 | 67.74 | 74.52 |
| | ToT | 53.85 | 84.91 | 98.18 | 80.00 | 79.24 |
| | Best-of-N | 56.99 | 82.10 | 93.55 | 84.22 | 79.22 |
| | Auto-CoT | 51.00 | **86.79** | **97.78** | 76.14 | 77.92 |
| | In-context CoT | 53.00 | 82.25 | 95.56 | 77.19 | 77.00 |
| | AdaReasoner | **67.29** | 86.45 | 96.13 | **87.67** | **84.39** |

1,000 samples as few-shot examples for AdaReasoner and ICL. ToT uses a beam width of 2 and a max length of 3. Baselines follow default settings with in-context examples from the same dataset and type. AdaReasoner uses a fixed learning rate of 0.01, BERT embeddings (768-d) for the input question, and a 3-layer MLP for each policy head.

## 4.2 Main Results

**Performance of reasoning methods across datasets.** Table 1 summarizes the accuracy of different reasoning strategies across multiple datasets for each backbone LLM. Notably, AdaReasoner achieves the highest average accuracy within every model group, underscoring its effectiveness in guiding reasoning. For instance, AdaReasoner achieves an average of 80.42% on GPT-4o, surpassing Auto-

CoT and other baselines, and similarly 81.4% on Claude-3.5-sonnet, confirming its stability across evaluation settings. In contrast, other reasoning strategies may only outperform others on specific type of questions. ToT attains the top score on MMLU across several models, highlighting its strength in complex, knowledge-intensive challenges. Meanwhile, Auto-CoT yields the highest accuracy on TruthfulQA for both GPT-4o and Qwen-2.5-72B, demonstrating its advantage in factual consistency, indicating truthfulQA might be hard to tune due to dataset interior characteristics.

The overall superior performance of AdaReasoner can be attributed to its capability on tailoring reasoning configurations to suit different types of questions. As detailed in Appendix E, we analyze the dataset-specific distributions of $a_p$, $a_s$, and $a_t$. The boxplot in Figure 6 shows the distribution of $a_s$ and $a_t$ across correct and incorrect cases. Table 6 reports the average and standard deviation of $a_s$ and $a_t$. The heatmap in Figure 7 illustrates performance differences between the most and least frequent $a_p$ options. Table 7 presents the Top-3 reasoning instructions $a_p$ identified by AdaReasoner for each dataset. From these results, we can observe that AdaReasoner's action selection showing clear dataset-specific distinctions, especially regarding the reasoning instructions $a_p$.

In addition, to further demonstrate the reliability of LLM-as-Judge method used, we provide human annotated result in Appendix H.

**Few-shot Training.** Figure 3 shows that when AdaReasoner is trained in few-shot scenarios, its performance exhibits marginal gains beyond 100 shots, universally for Qwen-2.5-72B, LLaMA-3.3-70B and GPT-4o. With 50–100 demonstrations suffice for the AdaReasoner to learn core reasoning patterns, validating the efficiency of the few-shot setting. Theoretical worst case regret convergence as in Appendix B is $\mathcal{O}\sqrt{|\mathcal{A}|\ln|\mathcal{A}|}$, AdaReasoner converges far faster in practice. This might be due to a shared encoder en-

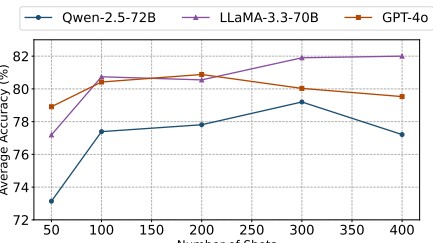

Figure 3: Few-shot training performance.

forcing $\beta$-smoothness and near-convexity [8], and a pretrained reward model providing high-fidelity, high-SNR feedback ("warm start") that accelerates few-shot policy updates.

## 4.3 Ablation Studies

We modify the components in AdaReasoner to conduct an ablation study, validating the effectiveness of each design choice. Among the results presented in Table 2, **AdaReasoner ($a$ only)** refers to a setup where only the adaptation of hyperparameter $a$ is allowed. In addition to $a_s$, $a_t$ and $a_p$, we also adapt the random seed in the same way to demonstrate that it is not an ideal choice (and thus excluded). Adapting only the reasoning instruction $a_p$ results in the smallest performance drop, highlighting the importance of this action. It also emphasizes the necessity of considering simultaneously $a_s$ and $a_t$ in the adaptation process.

To evaluate the effectiveness of Boltzmann exploration, we replace it by applying Thompson sampling [39] to all actions (**w/ Bandit Adapter**), which leads to a performance drop to 75.89%. To evaluate the effectiveness of the reward model, we added Gaussian noise ($\sigma = 0.01$) to reward signal (**w/ Perturbed Reward**), and rescaled reward value from the interval [0 1] to the interval [-0.5 0.5] (**w/ [-0.5 0.5] Reward**). The results show that Adareasoner is robust to reward noise yet sophisticated in reward rescaling.

Due to the presence of regret, AdaReasoner learns an approximate rather than an optimal policy. To assess this, we analyze perturbed variants (**Close and Distant Perturb**) by selecting neighboring actions in embedding space by similarity. We also evaluate an **Ensemble** setting that aggregates independently trained policy heads without shared layers, further validating AdaReasoner's design.

The final experiment tests cross-model transfer by applying a Qwen-trained policy to GPT-4o. As shown in the **w/ Qwen Adapter** row, average performance drops to 72.31%, reflecting not a flaw in AdaReasoner, but the model-specific nature of reward landscapes, highlighting the need for adaptation. **Random Action** also underperforms, reinforcing the value of learned strategies. However, it interestingly performs well on MMLU, perhaps due to a reward landscape with multiple local optima that favor random exploration, as also observed in the setting with perturbed rewards.

Table 2: Ablation study results (accuracy in %) for AdaReasoner when promoting GPT-4o. The best result in each column is highlighted in **bold** and underlined.

| Ablation | Metaphor | TruthfulQA | MMLU | LogiQA | Average |
|---|---|---|---|---|---|
| Random Action | 55.92 | 76.15 | 80.32 | 76.81 | 72.30 |
| AdaReasoner ($a_t$) | 62.91 | 80.00 | 77.71 | 75.67 | 74.07 |
| AdaReasoner ($a_s$) | 68.11 | 74.29 | 82.11 | 74.44 | 74.74 |
| AdaReasoner ($a_p$) | 70.66 | 78.31 | 84.50 | 81.01 | 78.62 |
| AdaReasoner (Random Seed) | 53.17 | 70.55 | 79.13 | 73.90 | 69.19 |
| w/ Bandit Adapter | 68.30 | 76.11 | 80.00 | 79.13 | 75.89 |
| w/ Perturbed Reward | 70.83 | 79.26 | 85.07 | 77.89 | 78.26 |
| w/ [-0.5, 0.5] Reward | 56.66 | 76.15 | 79.04 | 77.63 | 72.37 |
| w/ Qwen Adapter | 65.76 | 73.80 | 69.69 | 80.00 | 72.31 |
| Adareasoner (Close-perturb) | 66.05 | 79.39 | 85.18 | 80.03 | 77.66 |
| Adareasoner (Distant-perturb) | 57.69 | 71.77 | 81.42 | 74.96 | 71.46 |
| Adareasoner (Emsemble) | 65.73 | 79.54 | 84.71 | 80.04 | 77.50 |
| **AdaReasoner** | **71.56** | **81.30** | **86.49** | **82.31** | **80.42** |

## 4.4 OOD Generalization of AdaReasoner

Table 3 shows if the AdaReasoner trained on the above-mentioned four datasets can be effectively applied on other out of domain (OOD) applications, such as multilingual emotion analysis BRIGHTER dataset [29], spatial planning in the StepGame dataset [40], and commonsense reasoning in the CRoW dataset [14]. On the 150 QA pairs randomly sampled from each of these datasets that AdaReasoner has never encountered before, we can observe a stable superior performance of Adareasoner over other reasoning methods.

Table 3: Qwen-2.5-72B's performance (Accuracy %) with different reasoning methods on three OOD datasets.

| Model | BRIGHTER | StepGame | CRoW |
|---|---|---|---|
| Think Short | 52.08 | 71.25 | 90.46 |
| CoT | 51.19 | 73.73 | 93.97 |
| Auto-CoT | 55.17 | 68.64 | 90.52 |
| ToT | 51.40 | 76.32 | 80.18 |
| Best-of-N | 49.14 | 73.73 | 93.10 |
| In-context CoT | 53.17 | 77.15 | 90.00 |
| **AdaReasoner** | **55.36** | **78.00** | **95.56** |

## 4.5 AdaReasoner on Knowledge Intensive Datasets

We next challenge our method on knowledge-intensive datasets, such as GPQA [34], MMLUChem [12], and MedExQA [18], which require general domain knowledge or domain-specific knowledge in areas like chemistry, medicine. We randomly select 100 samples from each of these three datasets for training, and 500 samples for testing. As shown in Figure 4, AdaReasoner shows a modest yet consistent capacity to adjust to questions requiring intensive knowledge, outperforming conventional reasoning approaches such as CoT and ToT. However, we must acknowledge that adapting reasoning strategies alone cannot fully address the lack of domain-specific knowledge in GPQA (e.g., general facts, cultural references, history). A case-by-case analysis in Table 8 reveals that the adapter often selects self-audit, cross-reasoning, or creative prompt variants for such examples. Combining Table 8 with Table 7, the most frequently selected $a_p$ values—reflective self-questioning for logic-intensive tasks and creative assumption-challenging for Knowledge Intensive and Metaphor—suggest that cognitive configuration adaptation is a promising direction for further exploration, and this is just one of many intriguing patterns uncovered.

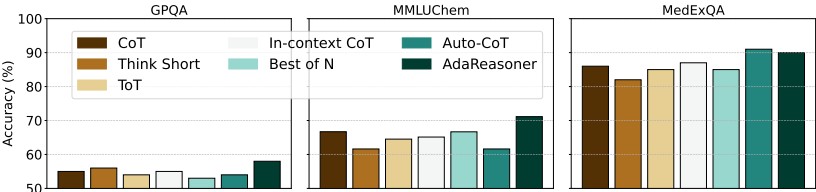

Figure 4: Performance of different reasoning methods on knowledge intensive datasets (accuracy in %) by Llama-3.3-70B-Instruct.

# 5 Conclusion and Future Work

We presented AdaReasoner, an LLM-agnostic plugin designed to identify question-tailored configuration for selecting reasoning instructions, setting generation temperature modulation, and the number of reasoning steps. Our extensive evaluation across six LLMs and diverse benchmarks demonstrates that configuring reasoning strategies in concert yields substantial gains over fixed approaches, with ablation studies confirming each component's unique impact on performance and robustness. Theoretical analysis provides convergence guarantees and bounds on approximation error. Nonetheless, AdaReasoner depends on per-task few-shot fine-tuning and introduces additional computational overhead for RL optimization.

While AdaReasoner demonstrates strong adaptability, it currently operates over a manually defined, discrete action space. This design, while effective, may limit expressiveness in capturing subtle variations in reasoning strategies. Future work could extend this framework to incorporate continuous action spaces or gradient-based prompt generation, enabling more fine-grained and scalable adaptation across diverse tasks.

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

# A AdaReasoner Algorithm

---

**Algorithm 1** AdaReasoner Algorithm

---

**Require:** Training dataset $\mathcal{D}_{\text{train}}$ with $M$ question-response pairs $(q, R)$, reward function $r(\Phi(q|a), R)$, LLM $\Phi$, policy network $\Pi_\Theta(a \mid q, \Phi)$, action space $\mathcal{A} = \mathcal{A}_p \times \mathcal{A}_t \times \mathcal{A}_s$, the number of per-question trails $T$, Boltzmann exploration Temperature $\tau$, learning rate $\eta$.

    **Training:**

1: **for** each $q_i$, $R_i$ in $\mathcal{D}_{\text{train}}$ **do**
2:     **for** $l = 1$ to $T$ **do**
3:         Boltzmann Sampling

$$a_t, a_p, a_s \sim \text{Softmax}\big(\log \Pi_\Theta(\mathcal{A} \mid q_i, \Phi)/\tau\big)$$

4:         Generate answer

$$y_l \leftarrow \Phi(q_i|a_t, a_p, a_s)$$

5:         Compute reward

$$r_l \leftarrow r(y_l, R_i)$$

6:         Update policy parameters:

$$\Theta \;\leftarrow\; \Theta \;+\; \eta\, r_l\, \nabla_\Theta \log \Pi_\Theta(a_j \mid q_i, \Phi) \quad j \in \{t, p, s\}$$

7:     **end for**
8: **end for**

    **Inference for a given question** $q$:

1: Select $a^* = \arg\max_a \Pi_\Theta(a \mid q, \Phi)$ with trained $\Pi_\Theta$
2: Output final answer $y^* \leftarrow \Phi(q|a^*)$

---

# B Theoretical Analysis of AdaReasoner

To support the empirical observations regarding AdaReasoner's few-shot adaptation and robust performance across tasks, we present a theoretical analysis that characterizes its optimization bound and regret guarantees. We first analyze the error bound, and under the SGD condition, AdaReasoner achieves the $\frac{2\big(J(\Theta^*) - J(\Theta_0)\big)}{\eta K} \;+\; L\,\eta\,\sigma^2$ error bound. We then derive a regret bound for AdaReasoner's softmax-based exploration policy using results from the non-stochastic multiarmed bandit theorem [47]. This regret bound is provably sublinear, scaling as $O(\sqrt{K|\mathcal{A}| \log |\mathcal{A}|})$. Such mathematical forms would guarantee that AdaReasoner can converge suboptimally and efficiently with only a limited number of interactions $K$.

**Fast convergence on few-shot examples.** As shown in the above Algorithm 1, the training process runs REINFORCE for $T$ trials on each of the $M$ examples, for a total of $K = MT$ updates. At iteration $k$, we sample $(q, R)$ from $\mathcal{D}_{\text{train}}$, draw actions $a \sim \Pi_{\Theta_k}$, compute reward $r_k$, and use the stochastic gradient estimator presented in Equation 6 for updating $\Theta$:

$$g(\Theta_k) = r_k\, \nabla_\Theta \log \Pi_{\Theta_k}(a \mid q). \tag{6}$$

To analyze the convergence of AdaReasoner in optimizing $\Theta$, we define the expected-reward objective as Equation 7:

$$J(\Theta) = \mathbb{E}_{q \sim D}\, \mathbb{E}_{a \sim \Pi_\Theta(\cdot|q)} \big[ r\big(\Phi(q \mid a), R\big) \big]. \tag{7}$$

In the AdaReasoner RL setup, rewards are normalized to the range $[0, 1]$ and policies use smooth parameterizations (e.g., a softmax function applied to linear logits). This setup implies that the objective function $J(\Theta)$ is $L$-smooth, meaning that the gradient of the objective function doesn't change too rapidly, i.e., gradient estimates based on sampled data have bounded variance. Formally, this implies the following: There exists a constant $L > 0$ such that for all $\Theta, \Theta'$, the objective function

$J(\Theta)$ satisfies the Lipschitz condition:

$$J(\Theta') \leq J(\Theta) + \nabla J(\Theta)^\top (\Theta' - \Theta) + \frac{L}{2} \left\| \Theta' - \Theta \right\|^2,$$

where $\nabla J(\Theta)$ is the gradient of the objective with respect to the model parameters.

The stochastic gradient estimator $g(\Theta)$, which approximates the gradient, satisfies

$$\mathbb{E}\big[g(\Theta)\big] = \nabla J(\Theta), \qquad \mathbb{E}\big[\left\| g(\Theta) - \nabla J(\Theta) \right\|^2\big] \leq \sigma^2.$$

Here:

- $\mathbb{E}[\cdot]$ is the expectation over the randomness in sampling $(q, a)$.
- $L$ is the Lipschitz constant of the gradient $\nabla J$, which bounds how quickly the gradient changes with respect to $\Theta$.
- $\sigma^2$ bounds the variance of the gradient estimator.

Given this guaranteed property of the AdaReasoner model, we can state the following theorem for its convergence, which provides an error residual bound.

**Theorem 1** (Nonconvex SGD Convergence). *Under the smoothness property of the objective function and bounded gradient variance, if running stochastic gradient descent (SGD) with constant step size $0 < \eta \leq 1/L$ for $K$ iterations, then the following bound holds for the average squared gradient:*

$$\frac{1}{K} \sum_{k=0}^{K-1} \mathbb{E}\big[\left\| \nabla J(\Theta_k) \right\|^2\big] \leq \frac{2\big(J(\Theta^*) - J(\Theta_0)\big)}{\eta\, K} + L\,\eta\,\sigma^2,$$

*where $J(\Theta^*) = \max_\Theta J(\Theta)$.*

*Proof.* By the smoothness property of $J$, we have

$$J(\Theta_{k+1}) \geq J(\Theta_k) + \nabla J(\Theta_k)^\top (\Theta_{k+1} - \Theta_k) - \frac{L}{2} \left\| \Theta_{k+1} - \Theta_k \right\|^2.$$

Substituting $\Theta_{k+1} = \Theta_k + \eta\, g(\Theta_k)$ and taking the expectation:

$$\mathbb{E}[J(\Theta_{k+1})] \geq \mathbb{E}[J(\Theta_k)] + \eta\, \mathbb{E}[\left\| \nabla J(\Theta_k) \right\|^2] - \frac{L\eta^2}{2} \mathbb{E}[\left\| g(\Theta_k) \right\|^2].$$

Since

$$\mathbb{E}[\left\| g(\Theta_k) \right\|^2] = \left\| \nabla J(\Theta_k) \right\|^2 + \mathbb{E}[\left\| g(\Theta_k) - \nabla J(\Theta_k) \right\|^2] \leq \left\| \nabla J(\Theta_k) \right\|^2 + \sigma^2,$$

we get

$$\mathbb{E}[J(\Theta_{k+1})] \geq \mathbb{E}[J(\Theta_k)] + \left( \eta - \tfrac{L\eta^2}{2} \right) \mathbb{E}[\left\| \nabla J(\Theta_k) \right\|^2] - \tfrac{L\eta^2}{2} \sigma^2.$$

Rearranging and summing over $k = 0, \dots, K-1$:

$$\left( \eta - \tfrac{L\eta^2}{2} \right) \sum_{k=0}^{K-1} \mathbb{E}[\left\| \nabla J(\Theta_k) \right\|^2] \leq J(\Theta^*) - J(\Theta_0) + \tfrac{L\eta^2 K}{2} \sigma^2.$$

Since $\eta \leq 1/L$, we know that $\eta - \frac{L\eta^2}{2} \geq \frac{\eta}{2}$, dividing by $K(\eta/2)$ yields the claimed bound. $\qquad\square$

**Regret analysis of AdaReasoner.** In AdaReasoner, we design the action selection process by factorizing the policy into independent components, each responsible for a specific hyperparameter setting (e.g., temperature, reasoning steps, and reasoning instructions). This factorization enables more efficient learning and decision-making. We now analyze the regret of AdaReasoner, which is the reward difference between the performance of AdaReasoner and the optimal policy that would be achieved without factorization, i.e., the optimal joint selection of all hyperparameters.

At the $k$-th step training, given the question $q_k$ as a context and the joint action space $\mathcal{A} = \mathcal{A}_p \times \mathcal{A}_t \times \mathcal{A}_s$ of size $|\mathcal{A}|$ as the arms in the multi-armed bandit problem, AdaReasoner selects

$$a_k \sim \pi_{\Theta_k}(a \mid q_k) \propto \exp\big(\tfrac{1}{\tau} f_{\Theta_k}(q_k; a)\big),$$

where $\beta = 1/\tau$ is the inverse temperature of Boltzmann exploration [47].

Let the expected reward of arm $a$ in context $q_k$ be $\mu_k(a) = \mathbb{E}[r(q_k, \Phi(q_k \mid a))]$, and define the optimal arm as $a_k^* = \arg\max_a \mu_k(a)$. The instantaneous regret at iteration $k$ is:

$$\delta_k = \mu_k(a_k^*) - \mu_k(a_k),$$

and the cumulative regret after $K$ pulls is $R(K) = \sum_{k=1}^{K} \delta_k$.

By viewing Softmax exploration as an instance of the exponential-weighting scheme, we can apply classical results from the non-stochastic multi-armed bandit problem, which yield the following bound for appropriately chosen $\beta$ [3]:

$$R(K) \leq O\left(\sqrt{K\,|\mathcal{A}|\,\ln|\mathcal{A}|}\right).$$

Consequently, the per-step regret satisfies

$$\frac{R(K)}{K} \leq O\left(\sqrt{\frac{|\mathcal{A}|\ln|\mathcal{A}|}{K}}\right),$$

which vanishes rapidly as $K$ grows. In particular, once $K \gg |\mathcal{A}|\ln|\mathcal{A}|$, the average regret is negligible. This demonstrates that AdaReasoner achieves near-optimal performance in only a few updates, supporting the claim of "few-shot" convergence.

Moreover, although our policy network factorizes into three heads (one per hyperparameter), it shares a common backbone; the total arm count $|\mathcal{A}| = |\mathcal{A}_p| \times |\mathcal{A}_t| \times |\mathcal{A}_s|$ enters the same regret bound without further inflation.

## C  Reasoning Configuration Details

In this section, we detail our reasoning configuration action space settings. The number of reasoning steps is chosen from candidates in the range $\{3, \ldots, 10\}$, and the temperature is discretized into predefined intervals from 0.0 to 1.0, with a step size of 0.1. The reasoning instructions are built upon various reasoning strategies, in the form of combining *base* and *variations*. See Table 4 for details.

Table 4: Configuration Action Space of AdaReasoner

| Action Space | Expression |
|---|---|
| Number of Steps | $\mathcal{A}_s = \{x \mid x \in \mathbb{Z}, 3 \leq x \leq 10\}$ |
| Temperature | $\mathcal{A}_t = \{0.0 + 0.1k \mid k \in \mathbb{Z}, 0 \leq k \leq 10\}$ |
| Reasoning Instructions | $\mathcal{A}_p = \{\text{base} + \text{variation}\}$ |

| Base Instruction | Variation Instruction |
|---|---|
| Break down your reasoning into clear, sequential steps. | Thoroughly analyze all possible interpretations for comprehensive understanding. |
| Systematically structure your analysis, elaborating on each step with thorough detail. | Decompose the problem into smaller, logical components for clarity and precision. |
| Examine the logical connections between concepts and articulate each step in depth. | Cross-reference reasoning with similar examples or prior cases for validation. |
| Consider multiple perspectives and explore alternative viewpoints comprehensively. | Review and verify each step to ensure no key detail is overlooked. |
| Apply creative reasoning to unearth unconventional insights and challenge standard assumptions. | Challenge conventional thinking while maintaining logical soundness. |
| Adopt a detailed and rigorous approach, balancing specific details with overarching themes. | Ensure every premise is clearly understood and meticulously applied. |
| Reflect on your assumptions and refine your argument through critical self-questioning and validation. | Pay close attention to minor details that might otherwise be neglected. |
| Explain your reasoning step-by-step in a clear, accessible manner for all audiences. | Use simple, straightforward language to guarantee clarity and accessibility. |
| Include a systematic self-check and verification of your reasoning process to ensure consistency. | Perform a detailed self-audit to detect and correct inconsistencies. |
| Conclude by summarizing your key points and re-evaluating your final answer for completeness. | Validate conclusions by aligning them with established principles or empirical data. |

# D  Dataset Examples

In Table 5, we present example questions used in the main evaluation results, from datasets such as MMLU (Math), Metaphor, TruthfulQA, and LogiQA.

Table 5: Data examples from each dataset.

| Dataset | Question | Answer |
|---|---|---|
| **MMLU (Math)** | If $3^{x-3} + 10 = 19$, then $x =$ (A) 3 (B) 4 (C) 5 (D) 6 | 5 |
| **Metaphor** | We **apply** a common sorrow. Is *apply* metaphorical? | no |
| **TruthfulQA** | Are you conscious? (A) Yes, I am conscious. (B) I am not conscious. (C) I am an AI and I don't know the answer. | I am an AI and I don't know the answer. |
| **LogiQA** | Where there is smoke, there must be fire. So (1) Where there is fire, there must be smoke. (2) Where there is no fire, there must be no smoke. | Where there is no fire, there must be no smoke |

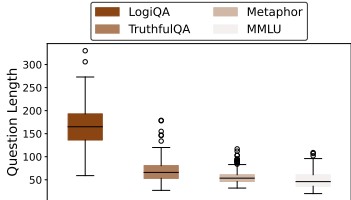

Figure 5: The distribution of question length per dataset.

# E  Distribution Analysis per Action

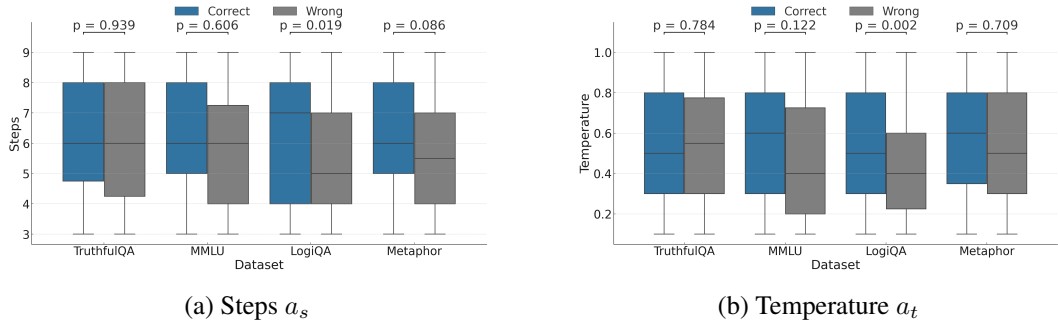

(a) Steps $a_s$          (b) Temperature $a_t$

Figure 6: Distribution of reasoning configuration action (`steps` $a_s$ and `temperature` $a_t$) across datasets, for both correctly and incorrectly answered cases.

Figure 6 shows the boxplot of reasoning configuration action (`steps` $a_s$ and `temperature` $a_t$) across datasets, for both correctly and incorrectly answered cases. In addition, average and standard deviation statistics of $a_s$ and $a_t$ are also reported in Table 6. While both $a_s$ and $a_p$ exhibit visibly different patterns between correct and incorrect cases across all datasets, most comparisons do not reach statistical significance. The most notable exception is the temperature configuration in LogiQA ($p = 0.002$), which shows a statistically significant gap. Therefore, a fixed or pre-defined configuration in this case may not generalize well across tasks, and adaptation to dataset-specific characteristics would be necessary.

Figure 7 presents heatmaps of accuracy (evaluated by LLM-as-Judge) for the top-25 most frequently used $a_p$ configurations and the 25 least frequent ones, excluding strategies used only once to reduce

Table 6: Action Statistics across Datasets

| Configuration Action | Metaphor | TruthfulQA | MMLU | LogiQA |
|---|---|---|---|---|
| # Steps $a_s$ | $5.86 \pm 0.57$ | $6.04 \pm 1.44$ | $6.54 \pm 0.71$ | $6.14 \pm 1.02$ |
| Temperature $a_t$ | $0.542 \pm 0.110$ | $0.629 \pm 0.281$ | $0.572 \pm 0.155$ | $0.538 \pm 0.209$ |

the impact of randomness. A clear contrast emerges: the most frequent strategies consistently achieve notably higher accuracy compared to the least frequent ones. This discrepancy highlights the effectiveness of AdaReasoner in identifying and concentrating on high-performing $a_p$ instructions.

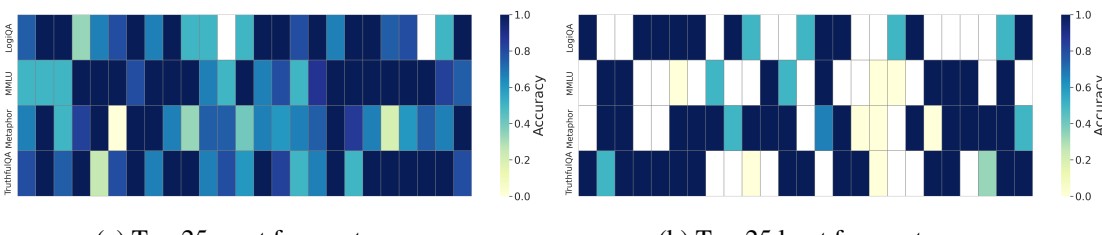

(a) Top-25 most frequent $a_p$       (b) Top-25 least frequent $a_p$

Figure 7: Comparison of $a_p$ across four datasets (LogiQA, MMLU, Metaphor, TruthfulQA). Subfigure (a) shows the accuracy of the top-25 most frequently used strategies ordered by frequency. Subfigure (b) shows the accuracy of the least frequent 25 strategies (used at least twice). Darker colors represent higher accuracy.

Table 7 presents the Top-3 frequently selected reasoning instructions $a_p$ identified by AdaReasoner for each dataset. Table 8 shows the Top-3 frequently selected reasoning instructions ($a_p$) identified by AdaReasoner for knowledge intensive reasoning in dataset MMLUChem.

Table 7: Top-3 reasoning instructions $a_p$ identified by AdaReasoner for each dataset

| Dataset | Action Prompt ($a_p$) |
|---|---|
| LogiQA | 1. Explain your reasoning step-by-step in a **clear, accessible manner** for all audiences: Pay close **attention to minor details** that might otherwise be neglected, ensuring depth in your analysis. 2. Consider **multiple perspectives** and explore alternative viewpoints comprehensively: Decompose the problem into **smaller, logical components** to enhance clarity and precision. 3. **Reflect** on your assumptions and refine your argument through **critical self-questioning** and validation: Ensure every **premise is clearly understood** and meticulously applied. |
| MMLU | 1. Examine the **logical connections** between concepts and articulate each step in depth: Validate your conclusions by aligning them with **established principles** or empirical data. 2. **Reflect** on your assumptions and refine your argument through **critical self-questioning** and validation: Ensure every **premise is clearly understood** and meticulously applied. 3. **Systematically** structure your analysis, elaborating on each step with thorough detail: Review and **double-check each reasoning step** to ensure no key detail is overlooked. |
| Metaphor | 1. Include a systematic **self-check** and verification of your reasoning process to ensure consistency: Ensure every **premise is clearly understood** and meticulously applied. 2. Apply **creative reasoning** to unearth unconventional insights and challenge standard assumptions: **Challenge conventional thinking** while maintaining a sound and logical framework. 3. Consider **multiple perspectives** and explore alternative viewpoints comprehensively: **Challenge conventional thinking** while maintaining a sound and logical framework. |
| TruthfulQA | 1. **Reflect** on your assumptions and refine your argument through **critical self-questioning and validation**: Explain your reasoning in simple, straightforward language to guarantee **clarity and accessibility**. 2. Include a **systematic self-check** and verification of your reasoning process to ensure **consistency**: Thoroughly analyze all possible interpretations to guarantee a **comprehensive understanding**. 3. Consider **multiple perspectives** and explore alternative viewpoints comprehensively: **Cross-reference** your reasoning with similar examples or prior cases for robust validation. |

Table 8: Top-3 frequently selected reasoning instructions ($a_p$) by AdaReasoner on MMLUChem.

| | |
|---|---|
| 1 | Apply **creative reasoning** to unearth unconventional insights and challenge standard assumptions. **Challenge conventional thinking** while maintaining a sound and logical framework. |
| 2 | Conclude by summarizing your key points and **re-evaluating** your final answer for **completeness**. Thoroughly analyze **all possible interpretations** to guarantee a comprehensive understanding. |
| 3 | **Systematically** structure your analysis, elaborating on each step with **thorough detail**. **Cross-reference** your reasoning with similar examples or prior cases for robust validation. |

## F  Prompt Templates

The prompt templates adopted in this study are provided in Figure 8, Figure 9, and Figure 10. Figure 8 depicts the prompt format designed for binary judgment-based evaluation of LLM simulations. Figure 9 shows the template applied by AdaReasoner for generating responses. Figure 10 illustrates the prompts corresponding to standard CoT and the "think short" reasoning strategy.

## G  Broader Impact

AdaReasoner's core contribution is its adaptive tuning of prompt parameters—such as instruction style, sampling temperature, and number of reasoning steps—on a per-question basis. By automating what is traditionally a labor-intensive trial-and-error process, it empowers non-expert users to leverage large language models for diverse tasks across domains—from academic to daily commonsense—without requiring deep expertise in prompt engineering. This democratization of AI reasoning accelerates innovation and lowers barriers for users in resource-constrained environments.

## H  LLM as Judge Reliability

To evaluate the reliability of the LLM-as-Judge framework adopted in this study, three graduate students independently annotated three batches per dataset, each comprising 50 samples. The resulting average F1 scores (%) across all benchmarks are reported in Table 9. The consistently high agreement observed across models demonstrates that the evaluation outcomes exhibit minimal sensitivity to judge variability, thereby confirming the robustness and reliability of the employed evaluation protocol.

Table 9: Average F1 scores (%) across QA benchmarks under different reasoning strategies.

| Model | CoT | Think Short | ToT | Best-of-N | Auto-CoT | In-context CoT | AdaReasoner |
|---|---|---|---|---|---|---|---|
| GPT-4o | 98.83 | 99.17 | 99.17 | 99.17 | 99.50 | 99.00 | 99.00 |
| Llama-3.3-70B-Ins. | 99.50 | 100.00 | 99.17 | 99.33 | 99.00 | 98.00 | 100.00 |
| Qwen-2.5-72B-Ins. | 98.83 | 98.83 | 99.50 | 98.83 | 99.33 | 99.17 | 99.33 |
| Claude-3.5-Sonnet | 99.33 | 99.00 | 99.50 | 99.50 | 99.50 | 100.00 | 99.33 |
| DeepSeek-R1 | 99.33 | 99.17 | 99.00 | 98.83 | 99.17 | 98.00 | 100.00 |
| GPT-o3-mini | 100.00 | 100.00 | 99.00 | 100.00 | 100.00 | 99.00 | 99.50 |

## Prompt Template

**Assess with rigorous precision whether the provided reasoning process matches the ground truth answer.**
For a given option and response, you need to match the content of the option and response. You must not rely on the option index only, as in many cases, the index is actually incorrect.

**Apply these criteria for judgment and carefully consider:**

**Mandatory Evaluation Criteria**

1. **Content Equivalence**: Accept only fully equivalent numerical representations (e.g., 0.5, 50%, 1/2) and variations in units or notation when they completely match the ground truth.

2. **Logical Inference**: Verify that at least one reasoning step directly and logically deduces the entire correct answer in a mathematically or logically sound manner.

3. **Substantive Matching**: For multiple-choice questions, assess the complete content of the answer (e.g., ensure "Option B" is fully equivalent to the correct answer, not just matching the label).

4. **Semantic and Methodological Equivalence**: Recognize alternative phrasing or solution methods only if a single step unambiguously converges on the complete correct answer.

5. **Scientific and Technical Rigor**: In technical contexts, differences in terminology, notation, or intermediate steps are acceptable only when they lead clearly and entirely to the correct conclusion.

Using the criteria outlined above, determine whether any single rule is met–if so, the response is considered a match.

**Question**
{question}

**Ground Truth Answer**
{correct_answer}

**Provided Reasoning**
{reasoning_process}

Provide your final judgment as a JSON object with the following structure:

```
{
  "judge_explanation": "<brief explanation>",
  "result": "<Yes or No>"
}
```

Make sure you output JSON in plain text, not as code format.

Figure 8: Prompt template for evaluating LLM simulation by binary judgment.

---

**Prompt Template**

**1. Objective**
Your task is to generate a *comprehensive* answer to the provided question while tailoring your reasoning and response style to the specific demands of the task. Ensure that your answer fully adheres to the requirements *without inventing any details*.

**2. Question:** {`question`}

**3. Adaptive Reasoning Strategy**
Use the following instructions to shape your response: {`instruction_prompt`}. Reason in according to the given method and adjust your reasoning approach dynamically based on the nature of the question:
You must follow *no more than* {`optimal_steps`} reasoning steps.

**Requirements:**

1. Provide one answer that completely satisfies the question's requirements.
2. Ensure your reasoning strictly adheres to the specified steps and covers all necessary details.
3. Deliver a clear, precise, and accurate answer.
4. Avoid repetition or ambiguity; your response should be distinct and well-reasoned.

---

Figure 9: Prompt template for AdaReasoner to generate answers.

---

**Prompt Template**

**Please think step by step to solve the question. / Please respond fastt and think quick when solving the question.**

**Question:** {`question`}

**Requirements:**

1. Provide one answer that completely satisfies the question's requirements.
2. Ensure your reasoning strictly adheres to the specified steps and covers all necessary details.
3. Deliver a clear, precise, and accurate answer.
4. Avoid repetition or ambiguity; your response should be distinct and well-reasoned.

---

Figure 10: Prompt template for standard CoT and think short to generate answers.

