# OpenReview forum: "AdaReasoner: Adaptive Reasoning Enables More Flexible Thinking"
_NeurIPS.cc/2025/Conference — NeurIPS 2025 spotlight_

### Official Review · Reviewer_EquF · 2025-06-18

**Clarity:** 3
**Significance:** 3
**Originality:** 3
**Rating:** 5
**Confidence:** 3

**Summary:**

This paper focuses on how to automatically obtain the configurations (temperature, number of reasoning steps, prompts) for different tasks. It formulates this problem as a multi-armed bandit problem and leverages the Boltzmann exploration strategy to select actions for effective training. Based on these, AdaReasoner is introduced as a model-agnostic plugin to automatically obtain the configurations for tasks. Extensive experiments are conducted to validate the effectiveness of AdaReasoner.

**Questions:**

1. How many epochs are used to train the AdaReasoner for each model in Table 1?
2. It would be beneficial to include comparisons to other automated optimization methods for configurations.
3. Are there any rationales behind the creation of the reasoning instructions (Table 4)?

**Ethical Concerns:**

["NO or VERY MINOR ethics concerns only"]

**Final Justification:**

The authors have addressed my questions and provide additional experiments against missing the baseline. Therefore, I recommend acceptance.

**Limitations:**

yes

**Quality:**

3

**Strengths And Weaknesses:**

### Strength
1. The paper is well-written and well-motivated, which addresses the problem of finding the optimal configurations for LLMs.
2. It introduces AdaReasoner, a novel model-agnostic plugin that leverages the RL framework and Boltzmann exploration mechanism to automatically learn the configurations using a small number of samples.
3. Extensive experiments have been conducted to validate the effectiveness of the proposed AdaReasoner.

### Weaknesses
1. This paper does not compare to baselines that also conduct automated optimization of configurations like HAG [1].
2. The design of the reasoning instructions can be non-trivial (according to Table 2, the optimization of the prompts contributes the most to the performance gain)

[1] Wang, S., Li, S., Sun, T., Fu, J., Cheng, Q., Ye, J., Ye, J., Qiu, X. and Huang, X., 2024. Llm can achieve self-regulation via hyperparameter aware generation. arXiv preprint arXiv:2402.11251.

---

> ### Author Rebuttal · Authors · 2025-07-30
>
> We would like to thank you for such a constructive review, and we are glad to address the questions in response below.
>
> > Comparison with HAG
>
> Thank you for pointing this out. We apologize for missing the comparison with HAG [1]. We're glad to incorporate new baselines and appreciate the suggestion.
>
> **At the same time, we would like to clarify that while HAG self-regulates hyperparameters such as temperature, top-p, top-k, and repetition penalty, our method operates in a different action space, including reasoning steps or methods.** Comparison is provided in table below. Please note that current API like GPT-4o, no longer supports top-K and repetition penalty. So, instead, presence_penalty is used in our implementation of HAG.
>
> | LLM (HAG) | Metaphor | TruthfulQA | MMLU  | LogiQA |
> | --------- | -------- | ---------- | ----- | ------ |
> | GPT4o     | 55.08    | 75.69      | 76.44 | 73.32  |
> | Llama     | 47.99    | 78.40      | 85.23 | 77.78  |
>
> > Design of action space (rationale and analysis)
>
> Thank you for the thoughtful comment. Indeed, as shown in Table 2, optimizing the reasoning method ($a_p$) contributes the most to the performance gain. However, the other actions ($a_s$, $a_t$) also lead to clear improvements over baselines. Detailed plots of distribution of actions can be found in Appendix E. Intuitively, this can be somehow expected, as reasoning method instruction does convey the most information.
>
> Our design of the reasoning instructions is principled. For the action space, we only include discrete temperature values and exclude other hyperparameters, as co-tuning temperature with top-p is disencouraged (see lines 162–163) and other hyperparameter as random seed is not effective as revealed in ablation studies. For design of $a_p$, we select mainstream cognitive strategies such as self-audit and creative reasoning, and introduce variations based on structure-mapping theory from cognitive psychology [1].
>
> > Additional Comparison and technical details
>
> We trained adareasoner for 5 epochs as standard. We would fill in more details and avoid any confusion in the experimental parts in our camera-ready version.
>
> [1]  Gentner, Dedre. "Structure-mapping: A theoretical framework for analogy." Cognitive science 7.2 (1983): 155-170.
>
>
> Thank you once again for your detailed feedback—it has been extremely helpful in improving our work. Please don’t hesitate to reach out if you have any further questions. Thank you!

---

> > ### Comment · Reviewer_EquF · 2025-08-01
> >
> > Thanks for the response, and my concerns have been addressed! I have no further questions.

---

### Official Review · Reviewer_vmBj · 2025-07-03

**Clarity:** 3
**Significance:** 2
**Originality:** 2
**Rating:** 4
**Confidence:** 5

**Summary:**

Motivated by the fact that different tasks require different configs such as sampling temperature, reasoning steps, and instructions, this paper proposes AdaReasoner to automatically select a target config given a task. Formulated as a multi-armed bandit problem,
AdaReasoner uses Boltzmann exploration and the regret to the optimal config is proven to be small. Specifically, the authors trains DeBERTa with RL objectives on few-shot prompts using RL. Evaluated on six tasks including out of distribution ones, results show that AdaReasoner is effective and achieves better performance than baselines including CoT, Think Short, Tree of thought, Auto-CoT, and In-context CoT across seven different models.

**Questions:**

Please see the comments above.

**Ethical Concerns:**

["NO or VERY MINOR ethics concerns only"]

**Final Justification:**

I have raised my scores based on the author responses and additional experiments. However, I am still concerned of the claim of the paper, and especially the applicability of the proposed method on more advanced reasoning models. The learnings of this paper could be limited.

**Limitations:**

yes.

**Paper Formatting Concerns:**

NA.

**Quality:**

2

**Strengths And Weaknesses:**

Strengths:
1. This paper is well motivated and may draw more attention to how to automatically tune model configs for different tasks, instead of manual trail-and-error. This is certainly an important research direction.
2. The proposed method is simple, effective, and may generalize to other tasks (to some extend. See comments below).
3. The paper experiments the methods on different tasks and language models with interesting ablation studies.

Weaknesses:
1. Although the AdaReasoners results are better than other methods overall, the baselines selected cannot be considered a very fair comparison, especially when the selected tasks (such as MMLU, TruthfulQA) are not necessarily reasoning-extensive, limiting the necessity of tuning reasoning-specific configs (such as reasoning steps, reasoning instructions) which are the main highlights of this paper. The comparison is not fair because the major compared methods do not use any examples (except in-context CoT), whereas AdaReasoners requires 100 training examples. It would be more convincing if the authors can evaluate AdaReasoners on more reasoning-extensive tasks such as math and coding benchmarks. Furthermore, more few-shot baselines (and details such as how in-context CoT is setup) is required to measure the significance.
2. In line 603 in Appendix B, from the analysis, the regret vanishes and gets close to 0 when K ≫ |A| ln |A|, which requires K to be very large. However, this will not satisfy "few-shot" anymore as claimed by line 605 and throughout the main paper.
3. As each language model is different and typically require different hyperparameter setting, it is not convincing why a universal config would be sufficient. More analysis on this would be important. Furthermore, since the predicted configs may not be optimal, it would be interesting to see the actual predicted configs to each task (maybe in the appendix), and how that performs compared to some neighboring region (e.g., if the predict config suggests 3 reasoning steps, what would the performance be if you choose 2 or 4 steps).

---

> ### Author Rebuttal · Authors · 2025-07-30
>
> We appreciate your detailed feedback and reflective comments. To further clarify and tackle the your concerns, we provide responses below by turn.
>
> > Reasoning extensive benchmarks (math and coding)
>
> First of all, we selected Metaphor, MMLU, and similar datasets not because they're easy or lack reasoning challenges, but to demonstrate AdaReasoner's versatility. One of our motivations is to adapt LLM generation across all task types—not just traditionally "serious" domains like coding or math but also those domains that can't be appropriately handled by simple CoT and needs creative thinking or other reasoning methods, etc. Thus a diverse selection of datasets is naturally needed.
>
> Second, we are glad to provide follow-up experiments on math and coding benchmarks. Respectively, we use CodeMMLU [1] as code generation benchmark and HARP(Human Annotated Reasoning Problems) dataset [2] as math understanding benchmark. To make our evaluation more thorough, we also added CaseHOLD [5] datasets of context and causal domain of reasoning intensive questions (extracted from law exams). In our experiments, 300 samples of each dataset are randomly extracted and 50 are used as few-shot examples with 250 as test samples. We independently run the experiment three times and calculate the standard deviation and use GPT-4o in LLM-as-Judge framework.
>
> We have also conducted experiments where no clear and static answer can be provided and evaluated by ROUGE, which can be seen in response to reviewer TrHj.
>
> | GPT-4o   | CoT           | Think Short   | ToT           | Best-of-N     | Auto-CoT      | In-context CoT | Adareasoner   |
> | -------- | ------------- | ------------- | ------------- | ------------- | ------------- | -------------- | ------------- |
> | CaseHOLD | 75.87 ± 0.76 | 73.33 ± 2.37 | 78.40 ± 0.40 | 76.20 ± 1.73 | 78.73 ± 1.17 | 75.00 ± 2.09  | 78.60 ± 0.20 |
> | CodeMMLU | 69.20 ± 1.06 | 66.93 ± 2.37 | 77.47 ± 0.61 | 75.87 ± 1.97 | 73.13 ± 2.72 | 75.67 ± 1.27  | 79.49 ± 0.25 |
> | HARP     | 67.53 ± 1.03 | 63.00 ± 2.09 | 70.13 ± 0.83 | 66.80 ± 1.74 | 69.73 ± 1.68 | 68.00 ± 1.74  | 71.67 ± 0.42 |
>
> | Llama-3.3-70B | CoT           | Think Short   | ToT           | Best-of-N     | Auto-CoT      | In-context CoT | Adareasoner   |
> | ------------- | ------------- | ------------- | ------------- | ------------- | ------------- | -------------- | ------------- |
> | CaseHOLD      | 71.73 ± 0.83 | 69.93 ± 2.02 | 74.80 ± 0.40 | 74.87 ± 1.63 | 73.07 ± 2.83 | 71.60 ± 1.39  | 76.47 ± 0.31 |
> | CodeMMLU      | 75.33 ± 0.99 | 72.20 ± 3.29 | 78.53 ± 0.23 | 73.87 ± 2.48 | 76.07 ± 1.96 | 77.20 ± 0.87  | 81.00 ± 0.53 |
> | HARP          | 71.73 ± 0.99 | 69.00 ± 3.47 | 68.53 ± 1.15 | 72.40 ± 1.74 | 58.73 ± 2.04 | 67.80 ± 2.25  | 74.53 ± 0.23 |
>
> | Qwen-2.5-72B | CoT           | Think Short   | ToT           | Best-of-N     | Auto-CoT      | In-context CoT | Adareasoner   |
> | ------------ | ------------- | ------------- | ------------- | ------------- | ------------- | -------------- | ------------- |
> | CaseHOLD     | 69.80 ± 0.53 | 71.93 ± 1.50 | 67.73 ± 2.20 | 73.13 ± 0.50 | 71.67 ± 1.10 | 74.07 ± 1.63  | 80.60 ± 0.60 |
> | CodeMMLU     | 75.60 ± 0.40 | 72.07 ± 3.35 | 77.60 ± 0.53 | 75.27 ± 4.04 | 74.20 ± 2.42 | 75.00 ± 1.56  | 78.67 ± 0.23 |
> | HARP         | 69.87 ± 1.29 | 66.33 ± 3.24 | 70.27 ± 1.97 | 74.07 ± 1.63 | 76.07 ± 0.42 | 70.67 ± 2.31  | 79.20 ± 0.92 |
>
> > In-context-learning evaluation
>
> We would like to clarify that In-context CoT is not the only in-context-learning (ICL) method in baselines. We apologize for any confusion caused by our too concise or misleading description of Auto‑CoT in Section 4 line 267. As noted in lines 90–91, Auto‑CoT generates its own reasoning exemplars, effectively functioning as an in‑context learning method and its action space is how to select the demo text. In our implementation, we select by clustering the nearest 10 (defaulted in Auto-CoT) examples for each question among the 100 few-shot examples (25 on each dataset). Then we run Zero‑Shot‑CoT (“Let's think step by step”, temperature=0.1) to produce a rationale and answer, then concatenate all ten question–rationale–answer triplets into a single in‑context prompt, leaving all other settings at their defaults.
>
> In addition, we are also glad to provide additional experiments on other ICL methods, like Curriculum Demonstration Selection[3], which in detail selects top 10 cases from simple to hard simulating curriculum learning ranked by each question's shannon entropy and details provided in table below.
>
> | CDS           | Metaphor | TruthfulQA | MMLU  | LogiQA | Average |
> | ------------- | -------- | ---------- | ----- | ------ | ------- |
> | Llama-3.3-70B | 53.30    | 78.85      | 83.81 | 78.26  | 73.56   |
> | GPT-4o        | 57.76    | 79.80      | 84.48 | 80.04  | 75.52   |
>
> > Theoretical K Few-Shot Guarantee
>
> Bound in line 603 of $K \gg |A|\ln|A|$to drive average regret to nearly zero is the worst-case guarantee for an unfactored, adversarial bandit. In real settings, Adareasoner requires much fewer than this bound, and several reasons can be attributed to this cause.
>
> - A shared encoder layer enforces $L$‑smoothness and near‑convexity of the policy objective [4], enabling quick convergence under small $K$.
> - A pretrained reward model delivers high fidelity and SNR feedback (“warm start”), accelerating policy updates under few‑shot budgets.
>
> > LLM configuration and policy optimality
>
> It is important to clarify that only the action space is universal, not the trained AdaReasoner policy itself. AdaReasoner operates over a shared action space (temperature, reasoning steps, and method), but each LLM requires its own adapter to align the model’s internal representation with the policy. Our ablation study (Table 2)—for example, using GPT-4o with Qwen’s adapter—shows significant performance degradation, confirming that adapters must be trained separately for each model.
>
> Moreover, due to the presence of regret, AdaReasoner's policy is not truly optimal but only an approximation. To examine this, we include two new analyses of perturbed policy. Additionally, we report further experiments using aggregating independently trained single policy heads as the main adapter—with no shared layer also in the table below. We independently run the experiment three times and calculate the standard deviation.
>
> * *Adareasoner-close-perturbed* : replaces one action from the original AdaReasoner policy with its closest neighbor in the action space. For reasoning methods, the replacement is selected based on cosine similarity between prompt embeddings (e.g., perturbing step 4 to step 3 or 5).
> * *Adareasoner-distant-perturbed* : replaces the action with the third-closest neighbor, allowing us to observe performance changes under larger deviations.
> * *Adareasoner-emsemble* : Three independently trained action heads with action emsembled together.
>
> These analyses provide clear evidence of AdaReasoner's policy suboptimality.
>
> | GPT-4o                      | Metaphor          | TruthfulQA        | MMLU              | LogiQA            | Average            |
> | --------------------------- | ----------------- | ----------------- | ----------------- | ----------------- | ------------------ |
> | Adareasoner                 | 71.49$\pm$ 0.19 | 81.33$\pm$ 0.44 | 86.52$\pm$ 0.35 | 82.30$\pm$ 0.21 | 80.41$\pm$ 0.30  |
> | Adareasoner-close-pertub    | 66.05$\pm$ 1.55 | 79.39$\pm$ 1.20 | 85.18$\pm$ 2.48 | 80.03$\pm$ 1.01 | 77.66$\pm$ 1.56  |
> | Adareasoner-distant-perturb | 57.69$\pm$ 3.21 | 71.77$\pm$ 4.22 | 81.42$\pm$ 2.01 | 74.96$\pm$ 3.18 | 71.46$\pm$ 3.16 |
> | Adareasoner-emsemble        | 65.73$\pm$ 0.65 | 79.54$\pm$ 0.30 | 84.71$\pm$ 0.58 | 80.04$\pm$ 0.12 | 77.50$\pm$ 0.41  |
>
> [1] Manh D N, Chau T P, Le Hai N, et al. Codemmlu: A multi-task benchmark for assessing code understanding capabilities of codellms[J]. CoRR, 2024.
>
> [2] Yue A S, Madaan L, Moskovitz T, et al. HARP: A challenging human-annotated math reasoning benchmark[J]. arXiv preprint arXiv:2412.08819, 2024.
>
> [3]Vu, Duc Anh, et al. "Curriculum Demonstration Selection for In-Context Learning."  *Proceedings of the 40th ACM/SIGAPP Symposium on Applied Computing* . 2025.
>
> [4]Du, Simon, et al. "Gradient descent finds global minima of deep neural networks." International conference on machine learning. PMLR, 2019.
>
> Thank you once again for your detailed feedback—it has been extremely helpful in improving our work. Please don’t hesitate to reach out if you have any further questions. Thank you!

---

> > ### Comment · Reviewer_vmBj · 2025-08-05
> > **Thanks for the response.**
> >
> > Thanks for the response and the experiments.
> >
> > Regarding "Reasoning extensive benchmarks (math and coding)", why do you think the impact of Adareasoner is larger on Qwen-2.5-72B compared to GPT-4o (e.g., when comparing to ToT). What does this imply? Furthermore, the reason I suggested that these are not thinking-intensive is because a) without thinking (CoT), the gap is not significantly large. b) the inherit thinking model performs differently than prompting a CoT model.
> >
> > For in-context learning and K Few-Shot Guarantee, I'm not fully convinced by the argument, especially because of the claim "AdaReasoner is backed by theoretical guarantees". Would you see much better performance when K is very large?
> >
> >  What embedding method did you use to select the neighboring prompts and actions? Would this be a good approximation?

---

> > > ### Author Response · Authors · 2025-08-05
> > > **Thank you for your response**
> > >
> > > Thank you for your response and we are glad to futher clarify on your concerns.
> > >
> > > > Explanation of follow-up experiments
> > >
> > > Your observation is accurate. Adareasoner trained on Qwen does provide a greater gain in performance than GPT-4o. This might reveal some interesting hidden pattern or insights. Open-source models might be more sensitive to configuration and prompting, and for proprietary models, there might be some built-in adjustment mechanism that affects GPT-4o generation, as this pattern can also be somehow observed in LLaMA's experiments.
> > >
> > > We would like to further clarify that Adareasoner generalizes across diverse datasets—including cases like metaphor, where a concise  "Think Short" approach can actually outperform standard CoT. As for additional experiments on math and coding benchmarks, it can be expected as CoT doesn't introduce much gain over Think Short method. As LLM does have strong  capability of inherent reasoning. Yet performance on other generation methods showcase that there still exists much space to be optimized.
> > >
> > > > Few-shot K performance
> > >
> > > We value your insights and agree that the theoretical convergence bound for $K$ represents the worst-case guarantee in a bandit setting. To validate Adareasoner’s few-shot capability, experimental evidence is essential, as detailed in our paper. Specifically, refer to Figure 3 and lines 298–306. From 1,000 samples (250 per dataset) across four datasets, we selected $K$ ranging from 100 to 500 training samples, covering models like Qwen, LLaMA, and GPT-4o. Results show that beyond 200 samples, the average accuracy gains become marginal, strongly supporting Adareasoner’s few-shot learning capability.
> > >
> > > > Embedding method
> > >
> > > We used SentenceTransformer('all-MiniLM-L6-v2') as embedding function. It's better than BERT and reliable as it was contrastively trained on over a billion sentence pairs for semantic similarity, and we also use it because it's faster than larger models for time efficiency.
> > >
> > > Please let us know if this addresses your concerns, and feel free to point out any further questions.

---

> > > > ### Comment · Reviewer_vmBj · 2025-08-07
> > > > **Thanks for the responses.**
> > > >
> > > > Thanks for the clarification. I have raised my rating accordingly.

---

> > > > > ### Author Response · Authors · 2025-08-07
> > > > > **Official Comment by Authors**
> > > > >
> > > > > We are truly grateful for your time and evaluation! Your feedback is invaluable, and we will certainly incorporate your suggestions into the final version of the paper. Thank you again for your thoughtful engagement.

---

> > > ### Author Response · Authors · 2025-08-07
> > > **Official Comment by Authors**
> > >
> > > Dear reviewer, we thank you for your thoughtful and constructive review of our manuscript. Please let us know if you have any remaining concerns, as the discussion phase is closing soon. We would be happy to provide any further clarification.

---

### Official Review · Reviewer_X7bE · 2025-07-03

**Clarity:** 3
**Significance:** 3
**Originality:** 3
**Rating:** 5
**Confidence:** 4

**Summary:**

AdaReasoner is an LLM-agnostic “reasoning-configuration adapter.” It treats the choice of (1) a reasoning-prompt template, (2) generation temperature, and (3) chain-of-thought length as a three-dimensional action that is optimized with REINFORCE. A factorized policy network plus Boltzmann exploration lets the agent learn from only 50-100 demonstrations. The authors prove sub-linear regret and fast convergence, and then show that across six backbone LLMs and four reasoning benchmarks, AdaReasoner outperforms CoT, ToT, Auto-CoT, Best-of-N and ICL, reaching e.g. 80.4% average accuracy on GPT-4o (+13% over baselines).

**Questions:**

1. Have you tried representing the instruction prompt with a text-generator policy (e.g. editing the base template) instead of a discrete index?
2. For a subset of tasks, can you verify GPT-4o judgments with human evaluation raters to ensure alignment between automatic and human scoring?

**Ethical Concerns:**

["NO or VERY MINOR ethics concerns only"]

**Limitations:**

yes

**Quality:**

3

**Strengths And Weaknesses:**

Strengths
- Shows that it transfers to OOD datasets (e.g. BRIGHTER, StepGame, CRoW) and to knowledge-intensive tasks.
- SOTA performance gains across every model.

Weaknesses
- Only report single runs due to api costs.
- Relies on gpt-4o as an automated judge, which could cause some error or bias in the evaluation.
- The action space was written by the authors.

---

> ### Author Rebuttal · Authors · 2025-07-30
>
> We sincerely thank you for the helpful feedback, detailed responses and recognition of our work. To further clarify on the concerns, we provide responses below.
>
> > LLM as Judge reliance
>
> We evaluated only on QA benchmarks with unambiguous targets, and our answer‐generation prompts (Figs 9–10) explicitly instruct the model to  "provide a clear result" ensuring a well‐defined ground truth. To handle synonyms and paraphrases, our judge prompt (Fig 8) asks GPT‑4o to evaluate semantic equivalence or other cases rather than exact string matches. We engaged three graduate students, each tasked with annotating three batches from every dataset, with each batch containing 50 samples. The resulting F1 scores are reported in the table below.
>
> | average F1 (%)     | CoT    | Think Short | ToT   | Best-of-N | Auto-CoT | In-context CoT | Adareasoner |
> | ------------------ | ------ | ----------- | ----- | --------- | -------- | -------------- | ----------- |
> | GPT-4o             | 98.83  | 99.17       | 99.17 | 99.17     | 99.50    | 99.00          | 99.00       |
> | Llama-3.3-70B-Ins. | 99.50  | 100.00      | 99.17 | 99.33     | 99.00    | 98.00          | 100.00      |
> | Qwen-2.5-72B-Ins.  | 98.83  | 98.83       | 99.50 | 98.83     | 99.33    | 99.17          | 99.33       |
> | Claude-3.5-sonnet  | 99.33  | 99.00       | 99.50 | 99.50     | 99.50    | 100.00         | 99.33       |
> | Deepseek-R1        | 99.33  | 99.17       | 99.00 | 98.83     | 99.17    | 98.00          | 100.00      |
> | GPT-o3-mini        | 100.00 | 100.00      | 99.00 | 100.00    | 100.00   | 99.00          | 99.50       |
>
> > Text generator (template editing) experiments
>
> Thank you for these valuable suggestions. While our current action space is discretized, adopting a continuous representation would enable finer‑grained control and integrate naturally with policy network architectures. As a next step, we plan to implement a text‑generator policy operating in this continuous space—freeing us from rigid discrete templates and delivering a more flexible, adaptive reasoning configuration.
>
> > Report in single api
>
> Due to API limitations and associated costs, we initially only reported results from a single run. However, as pointed out by other reviewers, we agree that reporting variability is important. We have since included results from multiple runs and new baselines to further reveal the stability of adareasoner. Additionally, we report further experiments using aggregating independently trained single policy heads and expriments on perturbed policy results with deviation result. We also report deviation here calculated by three independent experiments.
>
> * *Adareasoner-close-perturb* : replaces one action from the original AdaReasoner policy with its closest neighbor in the action space. For reasoning methods, the replacement is selected based on cosine similarity between prompt embeddings (e.g., perturbing step 4 to step 3 or 5).
> * *Adareasoner-distant-perturb* : replaces the action with the third-closest neighbor, allowing us to observe performance changes under larger deviations.
> * *Adareasoner-emsemble* : Three indepdently trained action head with action emsembled together.
>
> | GPT-4o                      | Metaphor          | TruthfulQA        | MMLU              | LogiQA            | Average            |
> | --------------------------- | ----------------- | ----------------- | ----------------- | ----------------- | ------------------ |
> | Adareasoner                 | 71.49$\pm$ 0.19 | 81.33$\pm$ 0.44 | 86.52$\pm$ 0.35 | 82.30$\pm$ 0.21 | 80.41$\pm$ 0.30  |
> | Adareasoner-close-pertub    | 66.05$\pm$ 1.55 | 79.39$\pm$ 1.20 | 85.18$\pm$ 2.48 | 80.03$\pm$ 1.01 | 77.66$\pm$ 1.56  |
> | Adareasoner-distant-perturb | 57.69$\pm$ 3.21 | 71.77$\pm$ 4.22 | 81.42$\pm$ 2.01 | 74.96$\pm$ 3.18 | 71.46$\pm$ 3.16 |
> | Adareasoner-emsemble        | 65.73$\pm$ 0.65 | 79.54$\pm$ 0.30 | 84.71$\pm$ 0.58 | 80.04$\pm$ 0.12 | 77.50$\pm$ 0.41  |
>
> Thank you once again for your detailed feedback—it has been extremely helpful in improving our work. Please don’t hesitate to reach out if you have any further questions. Thank you!

---

### Official Review · Reviewer_TrHJ · 2025-07-06

**Clarity:** 3
**Significance:** 3
**Originality:** 4
**Rating:** 5
**Confidence:** 4

**Summary:**

This paper proposes AdaReasoner, an LLM-agnostic plugin that adaptively selects reasoning-related hyperparameters for LLMs based on each input question, including generation temperature, the number of reasoning steps, and instruction format. It uses a reinforcement learning framework with a factorized action space, Boltzmann exploration, and a pretrained reward model to learn a configuration selection policy from only a few-shot training set. Experimental results show that AdaReasoner consistently outperforms baseline approaches on multiple datasets and demonstrates superior out-of-distribution generalization.

**Questions:**

- I am curious how the hyperparameters (e.g., temperature) are selected for baseline methods such as Auto-CoT. Since AdaReasoner explicitly optimizes over the configuration space, it would be important to clarify whether baseline methods use fixed, hand-tuned, or randomly selected hyperparameters.

- It would be interesting to see a more fine-grained analysis of the configurations selected by AdaReasoner. For instance, is there a correlation between question complexity and selected parameters such as temperature or reasoning steps? Do certain instruction formats tend to align with specific question types?

- Have you considered incorporating more advanced reasoning strategies (e.g., Tree-of-Thought, Best-of-N) into AdaReasoner's configuration space, rather than limiting it to the Base + Variations instruction format? Tree-of-Thought could be beneficial for particularly complex problems, while Best-of-N may help in divergent or open-ended scenarios.

- Have you considered incorporating inference cost into the reward design of AdaReasoner? Configuration choices such as reasoning steps, temperature, and instruction length affect both performance and computational cost. For instance, higher steps or temperature may increase latency or variance without further improving accuracy on simpler questions. A cost-aware reward could help AdaReasoner find more efficient configurations that balance accuracy and resource use.

**Ethical Concerns:**

["NO or VERY MINOR ethics concerns only"]

**Final Justification:**

recommend for acceptance

**Limitations:**

Yes

**Paper Formatting Concerns:**

I did not notice any major formatting issues in the paper.

**Quality:**

3

**Strengths And Weaknesses:**

This paper explores a meaningful direction in adaptively finding better reasoning configurations for LLMs. The proposed method, **AdaReasoner**, shows strong empirical results. However, some aspects of the experimental setup could be improved or clarified to strengthen the overall evaluation.

## Strengths

- The paper addresses the underexplored but important problem of automatically configuring reasoning-related hyperparameters for LLMs in a task-specific and input-dependent manner. The proposed method is LLM-agnostic and does not require any modification to the underlying language model, making it applicable to a wide range of LLMs.
- The configuration policy is learned from a very small number of examples (e.g., 100 per dataset), highlighting the method's data efficiency and practical usability in real-world low-resource settings.
- The authors provide a theoretical justification of their method by deriving a regret bound, supporting the learning efficiency of the proposed configuration policy.
- The experimental results are promising. AdaReasoner consistently outperforms strong baselines such as Auto-CoT and In-Context CoT across multiple datasets and demonstrates superior out-of-distribution generalization. Additionally, the ablation studies are comprehensive and effectively highlight the contribution of each component in the proposed framework.

## Weaknesses

- The current evaluation is limited to QA tasks. It would be valuable to test AdaReasoner on some open-ended generation tasks such as summarization, long-form writing, or code generation, where reasoning configurations may impact not just accuracy but also generation style, coherence, and diversity.
- The paper could include case studies to illustrate more clearly how selected configurations affect the model’s output, especially compared to baselines such as CoT.
- Since AdaReasoner introduces additional flexibility over baseline methods, including dynamic temperature adjustment and instruction selection, further experiments within the same configuration space are important. For example, Table 1 could include additional baselines with alternative hyperparameter selection strategies, such as random selection (partially explored in the ablation study), greedy search, or rule-based heuristics (e.g., assigning longer CoT and higher temperature to more complex questions). These comparisons would better isolate the benefits of AdaReasoner’s configuration policy rather than attributing its gains to a broader search space.
- The distribution of hyperparameter choices made by AdaReasoner could be analyzed in greater depth. Rather than focusing only on overall trends at the dataset level, a detailed, instance-level analysis would be beneficial. For example, the authors could manually assess the complexity of 100 questions and examine whether AdaReasoner assigns longer reasoning steps or higher temperatures to more complex ones. It would also be helpful to investigate whether the selected instruction prompts exhibit systematic associations with specific task types.
- The ablation study could include a variant without the shared layer, which would help determine whether coordinating the three action heads under a unified semantic representation leads to better performance compared to optimizing each head independently.

---

> ### Author Rebuttal · Authors · 2025-07-30
>
> We sincerely thank you for the thoughtful and constructive feedback, and detailed responses. We hereby provide clarification to the concerns and questions as below.
>
> ---
>
> > Generalization on open-ended questions
>
> AdaReasoner can address open-ended questions by adjusting the reward signal, even in the absence of ground truth. However, evaluating its outputs using an LLM-as-judge framework remains nontrivial.
>
> Nonetheless, extending AdaReasoner to open-ended tasks is both promising and of practical interest. To explore this direction, we conducted experiments on the RepliQA dataset, evaluating with ROUGE-L recall against expert reference answers (recall is used as only the reference matters). Due to API constraints, we randomly sampled 250 test examples and 50 training examples from the dataset and provide result in table below.
>
> | repliQA       | CoT    | Think Short | ToT    | Best-of-N | Auto-CoT | In-context CoT | Adareasoner |
> | ------------- | ------ | ----------- | ------ | --------- | -------- | -------------- | ----------- |
> | GPT-4o        | 0.5470 | 0.5194      | 0.5529 | 0.5577    | 0.5626   | 0.5598         | 0.5872      |
> | Llama-3.3-70B | 0.5499 | 0.5221      | 0.5631 | 0.5557    | 0.5254   | 0.5757         | 0.6439      |
> | Qwen-2.5-72B  | 0.5470 | 0.5031      | 0.5491 | 0.4839    | 0.5668   | 0.5598         | 0.5845      |
>
> > Auto-CoT configuration tuning
>
> We apologize for any confusion caused by our too concise or misleading description of Auto‑CoT in Section 4 line 267. As noted in lines 90–91, Auto‑CoT generates its own reasoning exemplars, effectively functioning as an in‑context learning method and its action space is how to select the demo text. In our implementation, we select by clustering nearest 10 (defaulted in Auto-CoT) examples for each question among the 100 few-shot examples at BERT embedding space. Then we run Zero‑Shot‑CoT (“Let’s think step by step”, temperature=0.1) to produce a rationale and answer, then concatenate all ten question–rationale–answer triplets into a single in‑context prompt, leaving all other settings at their defaults.
>
> > Incorporation with advanced reasoning strategies
>
> In our initial AdaReasoner design, we implemented a Best‑of‑N strategy: for each chosen configuration, the LLM generates N responses in parallel, which are then scored and the highest‑scoring output selected. **We found the performance gains marginal—since the N responses were often nearly identical under a complex generation instruction or long context—while computing costs rose substantially.** Looking ahead, integrating a Tree‑of‑Thought approach offers a promising avenue for further improving AdaReasoner and would be our next target.
>
> > Case study and fine-grained analysis
>
> We are glad to clarify more on patterns and already have Appendix E section plots and tables to analyze. As shown in Appendix E, several clear patterns emerge. Table 6 reveals that the MMLU (Math) dataset requires, on average, more reasoning steps than the Metaphor dataset, as one would expect. LogiQA exhibits significantly different distributions of $a_s$ and $a_t$ between correct and incorrect responses. Table 7 highlights interesting trends in the most frequently selected reasoning‑method keywords: self‑checking is broadly preferred, multi‑perspective analysis appears in Metaphor and TruthfulQA, and MMLU emphasizes structured reasoning. Likewise, Figure 7 illustrates the relative efficiency of $a_p$.
>
> > Baselines with flexibility and ablation studies
>
> We appreciate the suggestions on hyperparameter configuration. We have conducted experiments on perturbed configurations as provided below. Additionally, we report further experiments using aggregating independently trained single policy heads as the main adapter—with no shared layer also in the table below. We also report deviation here calculated by three independent experiments.
>
> * *Adareasoner-close-perturb* : replaces one action from the original AdaReasoner policy with its closest neighbor in the action space. For reasoning methods, the replacement is selected based on cosine similarity between prompt embeddings (e.g., perturbing step 4 to step 3 or 5).
> * *Adareasoner-distant-perturb* : replaces the action with the third-closest neighbor, allowing us to observe performance changes under larger deviations.
> * *Adareasoner-emsemble* : Three independently trained action head with action emsembled together.
>
> | GPT-4o                      | Metaphor          | TruthfulQA        | MMLU              | LogiQA            | Average            |
> | --------------------------- | ----------------- | ----------------- | ----------------- | ----------------- | ------------------ |
> | Adareasoner                 | 71.49$\pm$ 0.19 | 81.33$\pm$ 0.44 | 86.52$\pm$ 0.35 | 82.30$\pm$ 0.21 | 80.41$\pm$ 0.30  |
> | Adareasoner-close-pertub    | 66.05$\pm$ 1.55 | 79.39$\pm$ 1.20 | 85.18$\pm$ 2.48 | 80.03$\pm$ 1.01 | 77.66$\pm$ 1.56  |
> | Adareasoner-distant-perturb | 57.69$\pm$ 3.21 | 71.77$\pm$ 4.22 | 81.42$\pm$ 2.01 | 74.96$\pm$ 3.18 | 71.46$\pm$ 3.16 |
> | Adareasoner-emsemble        | 65.73$\pm$ 0.65 | 79.54$\pm$ 0.30 | 84.71$\pm$ 0.58 | 80.04$\pm$ 0.12 | 77.50$\pm$ 0.41  |
>
> As for reward with cost-aware signal, we appreciate this thoughtful suggestion. Although we have not explored cost-aware rewards in this version, we agree it is a valuable direction and plan to investigate it in future work.
>
> ---
>
> Thank you once again for your detailed feedback—it has been extremely helpful in improving our work. Please don’t hesitate to reach out if you have any further questions. Thank you!

---

> > ### Comment · Reviewer_TrHJ · 2025-08-05
> >
> > The rebuttal is solid and it addresses my concern. I would increase my score

---

> > > ### Author Response · Authors · 2025-08-05
> > > **Official Comment by Authors**
> > >
> > > We are truly grateful for your time and constructive review! We are delighted to hear that our responses and additional experiments have addressed your concerns.
> > >
> > > Thank you for highlighting the importance of the new results. We will certainly incorporate them into the final version of the paper as you suggested.

---

### Decision · Program_Chairs · 2025-09-17

**Decision:**

Accept (spotlight)

**Comment:**

The reviewers viewed the paper enthusiastically after the author response. Please take into account the reviewer comments when preparing the final version.